# Acute Hemorrhagic Encephalomyelitis in the Context of MOG Antibody-Associated Disease. Comment on Chen et al. Rapid Progressive Fatal Acute Hemorrhagic Encephalomyelitis. *Diagnostics* 2023, *13*, 2481

**DOI:** 10.3390/diagnostics13193148

**Published:** 2023-10-07

**Authors:** Sohyeon Kim, Mi-Yeon Eun, Jae-Joon Lee, Hung Youl Seok

**Affiliations:** 1Department of Neurology, Dongsan Hospital, Keimyung University School of Medicine, Daegu 42601, Republic of Korea; bjksh@naver.com (S.K.); hypersonan@gmail.com (J.-J.L.); 2Department of Neurology, School of Medicine, Kyungpook National University Chilgok Hospital, Daegu 41404, Republic of Korea; eunmiyn@gmail.com

**Keywords:** acute hemorrhagic encephalomyelitis, myelin oligodendrocyte glycoprotein

## Abstract

The study by Chen et al. of a 56-year-old man diagnosed with acute hemorrhagic encephalomyelitis (AHEM) had a significant impact on us. The authors provided a comprehensive account of their diagnostic journey and emphasized the need to differentiate myelin oligodendrocyte glycoprotein antibody-associated disease (MOGAD) from AHEM. However, recent research suggests that AHEM may not be an isolated entity, but rather a phenotype within MOGAD. The patient’s clinical presentation included MRI brain lesions characteristic of MOGAD in addition to hemorrhagic abnormalities. These findings raise the possibility that AHEM in this case represents a MOGAD phenotype. In conclusion, it is important to recognize the potential association between AHEM and MOGAD, especially when distinct MOGAD brain MRI patterns are present, as in this case.

Acute hemorrhagic encephalomyelitis (AHEM), also known as acute hemorrhagic leukoencephalitis or Hurst disease, is an extremely rare demyelinating disorder characterized by distinctive clinical features, diagnostic complexity, and limited treatment options [1,2,3]. This disease has a predilection for affecting adults, particularly males, more frequently than children [1,2,3]. It typically presents suddenly with hyperacute or acute symptoms, leading to rapid neurological deterioration with severe manifestations, including altered consciousness, seizures, and, in some cases, a coma [1,3]. The underlying cause often eludes identification, although some patients have evidence of antecedent or concurrent infections (e.g., Epstein–Barr virus, mumps, varicella-zoster virus, herpes simplex virus, human herpesvirus 6, and influenza) or the presence of autoantibodies (e.g., to myelin oligodendrocyte glycoprotein (MOG)) at the time of diagnosis or death [1,2,3,4,5,6,7]. The exact pathogenic mechanisms of AHEM remain uncertain, although the leading hypothesis revolves around an autoimmune response possibly triggered by viral or bacterial infections, perhaps due to molecular mimicry [1,2]. In addition, some studies have suggested the possibility of acute cerebral vasculitis induced by autoantibodies, such as MOG antibodies [5].

The diagnosis of AHEM depends on several factors, including clinical presentation, neuroimaging, cerebrospinal fluid (CSF) analysis, and, occasionally, histopathology [1,2]. Brain magnetic resonance imaging (MRI) plays a key role, showing characteristic multiple white matter lesions with edema and hemorrhage [1,2,3]. CSF analysis distinguishes AHEM from acute disseminated encephalomyelitis (ADEM), which typically shows lymphocyte-predominant mild to moderate pleocytosis [3]. In AHEM, neutrophilic pleocytosis is prominent and, importantly, red blood cells are occasionally observed [3]. Histopathologic findings may include a predominance of neutrophils in perivascular infiltrates, small vessel necrosis, and hemorrhage [1,2,3,5]. However, distinguishing AHEM from other neurological disorders, such as ADEM, infectious encephalitides, and other demyelinating diseases, can be a formidable challenge [2,3].

Due to its rarity, there is no established standard treatment protocol for AHEM. Typically, high-dose intravenous methylprednisolone is used as first-line treatment, with second-line interventions, such as plasma exchange and intravenous immunoglobulin, considered if intravenous methylprednisolone proves ineffective [3]. Outcomes vary widely among individuals, with mortality rates ranging from 46.5% to 70% [1,2,3]. However, some patients experience complete recovery or minimal neurological deficits [2,3].

We were very impressed by the study by Chen et al., which focused on a 56-year-old male patient diagnosed with AHEM [8]. The patient presented with rapid and progressive mental changes, sensory aphasia, right facial palsy, and right hemiparesis. Although histologic examination was not performed, MRI of the brain showed T2 hyperintense lesions with areas of hemorrhage in the supratentorial white matter and pons, leading to the diagnosis of AHEM. Unfortunately, despite intensive immunotherapy with steroid pulses and plasmapheresis, the patient died on day 35 of hospitalization. The authors meticulously detailed the process of distinguishing between various conditions in the diagnosis of this patient with AHEM. They particularly underscored MOG antibody-associated disease (MOGAD) as a distinct condition requiring differentiation from AHEM. However, our perspective on this issue is somewhat different. While we agree with the diagnosis of AHEM in this particular case, we are inclined to engage in a discussion about whether AHEM should be considered a completely separate entity from MOGAD or whether it could potentially manifest as a clinical presentation or phenotype of MOGAD. This is the focus of the discussion that we intend to explore further.

AHEM has conventionally been viewed as a severe form of ADEM [1]. However, based on our recent literature review, we propose that AHEM could potentially emerge as a phenotype of MOGAD [4]. To date, only three cases of MOG-IgG-associated AHEM have been documented [5,6,7], and the symptoms presented by the patient, including altered consciousness, aphasia, and hemiparesis, were previously identified as manifestations of MOG-IgG-associated AHEM [4]. Of these three cases, one involved a patient in their 50s who had a similar rapid progression to the patient in question and succumbed to the disease despite aggressive immunotherapy with corticosteroids, plasmapheresis, and cyclophosphamide [6]. This suggests that AHEM may not be an entirely distinct disease entity from MOGAD, but rather a clinical manifestation or phenotype thereof. As seen in this patient, MOG-IgG-associated AHEM can lead to rapidly progressive neurological impairment and a fatal outcome despite intensive immunotherapy. While the authors reported negative results for blood autoimmune autoantibodies, aquaporin-4 (AQP4), CSF oligoclonal bands, and CSF autoantibodies, they did not present the MOG antibody results. This raises the possibility that MOG antibody testing might not have been conducted.

Brain MRI findings in MOGAD typically show fluffy or poorly delineated T2 hyperintense lesions in various brain regions, including supratentorial and often infratentorial white matter, deep gray matter, middle cerebellar peduncle, large brainstem (pons and medulla), and confluent cortical areas [9,10]. Contrast enhancement patterns may be nonspecific, with leptomeningeal enhancement around the brainstem and linear leptomeningeal enhancement in cases of cerebral cortical encephalitis [9]. These lesions may partially or completely resolve over time, with infrequent silent lesion accrual and extremely rare residual T1 hypointense lesions, aiding in the diagnosis and evaluation of MOGAD [9].

The patient’s brain MRI showed extensive, poorly defined T2 hyperintense lesions involving both the frontal and parietal lobes, as well as a similarly large and ill-defined T2 lesion in the right middle cerebellar peduncle. These specific brain MRI findings are consistent with the common observations in MOGAD, which are characterized by bilateral, large, and ill-defined lesions, often involving the supratentorial white matter and deep gray matter [9,10]. The frequent involvement of the pons and the substantial lesions in the middle cerebellar peduncle in MOGAD patients further underscore this correlation [9]. Thus, the brain MRI findings in this patient provide a basis for considering the plausibility that AHEM in this particular case represents a phenotype of MOGAD.

The patient reported blurred vision two weeks prior to admission, although the authors did not provide a comprehensive description or documentation of this symptom. Nevertheless, it is important to consider whether this symptom could potentially be indicative of optic neuritis. If so, it would further support the idea of a possible association with MOG-IgG-associated AHEM.

The diagnostic criteria for MOGAD consist of three main components: (A) core clinical demyelinating events, including conditions such as optic neuritis, myelitis, ADEM, cerebral deficits, brainstem or cerebellar deficits, and cerebral cortical encephalitis with seizures; (B) positive MOG-IgG test results, which can be clear positive in serum without additional features, low positive in serum with supporting clinical or MRI features, or positive in CSF with supporting features and AQP4-IgG seronegative status; and (C) exclusion of better diagnoses, including multiple sclerosis [9]. According to these diagnostic criteria, if this patient, who has core clinical demyelinating events of brainstem deficit and MRI features supporting brainstem syndrome, tests positive for MOG antibodies, a diagnosis of MOGAD can be made, regardless of the antibody titer being high or low.

In conclusion, it is crucial to consider the potential association between AHEM and MOGAD, especially in patients with distinct MOGAD brain MRI patterns similar to the current case. Therefore, we strongly recommend MOG antibody testing. The exact contribution of MOG antibodies to the development of AHEM is not yet fully understood. However, one study has suggested a possible association between cerebral vasculitis associated with MOG antibodies and the onset of AHEM [5]. Further investigation is warranted to gain a more comprehensive understanding of this relationship.

## Data Availability

No new data were created or analyzed in this study.

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
