# Peer review of "Acute Hemorrhagic Encephalomyelitis in the Context of MOG Antibody-Associated Disease. Comment on Chen et al. Rapid Progressive Fatal Acute Hemorrhagic Encephalomyelitis. *Diagnostics* 2023, *13*, 2481"

_diagnostics, 2023, doi:10.3390/diagnostics13193148_

Round 1
Reviewer 1 Report
The authors comment on a recent case report presenting a case of AHEM. AHEM (also known as AHLE or Hurst disease) is a rare and often fatal acute encephalomyelitis. In many cases a cause has not been identified, although in some patients evidence of acute infection (e.g. EBV) or autoantibodies (e.g. against MOG) were present at time of diagnosis/death.
This comment is in between a comment on a single case report and a review of MOGAD vs. AHEM. I think this topic deserves more than a commentary and suggest that the authors transform it into a more comprehensive review on AHEM and its underlying causes.
Author Response
Thank you for considering our manuscript. We also thank the reviewers for their comments. We hope that we have adequately addressed all the concerns raised. A point-by-point response follows.
Reviewer #1
The authors comment on a recent case report presenting a case of AHEM. AHEM (also known as AHLE or Hurst disease) is a rare and often fatal acute encephalomyelitis. In many cases a cause has not been identified, although in some patients evidence of acute infection (e.g. EBV) or autoantibodies (e.g. against MOG) were present at time of diagnosis/death.
This comment is in between a comment on a single case report and a review of MOGAD vs. AHEM. I think this topic deserves more than a commentary and suggest that the authors transform it into a more comprehensive review on AHEM and its underlying causes.
Response: We value your feedback and have carefully considered your suggestions to improve the manuscript. In response to your recommendations, we have expanded the manuscript to provide a more comprehensive review of AHEM and its underlying causes. Below you'll find the additional content that has been seamlessly integrated into the manuscript: “Acute hemorrhagic encephalomyelitis (AHEM), also known as acute hemorrhagic leukoencephalitis or Hurst disease, is an extremely rare demyelinating disorder characterized by distinctive clinical features, diagnostic complexity, and limited treatment options. This disease has a predilection for affecting adults, particularly males, more frequently than children. It typically presents suddenly with hyperacute or acute symptoms, leading to rapid neurological deterioration with severe manifestations, including altered consciousness, seizures, and in some cases, coma. The underlying cause often eludes identification, although some patients have evidence of antecedent or concurrent infections (e.g., Epstein-Barr virus, mumps, varicella-zoster virus, herpes simplex virus, human herpesvirus 6, and influenza) or the presence of autoantibodies (e.g., to myelin oligodendrocyte glycoprotein (MOG)) at the time of diagnosis or death. The exact pathogenic mechanisms of AHEM remain uncertain, although the leading hypothesis revolves around an autoimmune response possibly triggered by viral or bacterial infections, perhaps due to molecular mimicry. In addition, some studies have suggested the possibility of acute cerebral vasculitis induced by autoantibodies such as MOG antibodies.
The diagnosis of AHEM depends on several factors, including clinical presentation, neuroimaging, cerebrospinal fluid (CSF) analysis, and occasionally histopathology. Brain magnetic resonance imaging (MRI) plays a key role, showing characteristic multiple white matter lesions with edema and hemorrhage. CSF analysis distinguishes AHEM from acute disseminated encephalomyelitis (ADEM), which typically shows lymphocyte-predominant mild to moderate pleocytosis. In AHEM, neutrophilic pleocytosis is prominent and, importantly, red blood cells are occasionally observed. Histopathologic findings may include a predominance of neutrophils in perivascular infiltrates, small vessel necrosis, and hemorrhage. However, distinguishing AHEM from other neurological disorders such as ADEM, infectious encephalitides, and other demyelinating diseases can be a formidable challenge.
Due to its rarity, there is no established standard treatment protocol for AHEM. Typically, high-dose intravenous methylprednisolone is used as first-line treatment, with second-line interventions such as plasma exchange and intravenous immunoglobulin considered if intravenous methylprednisolone proves ineffective. Outcomes vary widely among individuals, with mortality rates ranging from 46.5% to 70%. However, some patients experience complete recovery or minimal neurological deficits.”
We hope that these improvements meet your expectations and improve the completeness of the manuscript. Thank you very much for your valuable input.

Reviewer 2 Report
Minor changes in the attached file
As the possibility of an hemorragic MOGAD wasn't poposed by Chen et al. I suggest the authors to better contestualize their hypotesis, by providing a quick background on anti-MOG disease, through a better explanation of the possible MRI appearance of the disease (I suggest a reference on this topic in the attached file) and diagnostic criteria

Author Response
Thank you for considering our manuscript. We also thank the reviewers for their comments. We hope that we have adequately addressed all the concerns raised. A point-by-point response follows.
Reviewer #2
- Minor changes in the attached file.
1) I don't know if according to the journal guidelines, a commentary could have a different title, without any explicit citation to the article that is commented or a shorter one.
Response: We sincerely appreciate your insightful feedback. As a result of your input, we have changed the title to: “Acute Hemorrhagic Encephalomyelitis in the Context of MOG 2 Antibody-Associated Disease”.
2) We were very impressed by the study by Chen et al. which (→ who) focused on a 56-year-old 15 male patient diagnosed with acute hemorrhagic encephalomyelitis (AHEM).
Response: We highly value your insightful feedback. Regarding the latter sentence, since it pertains to Chen et al.'s study rather than the authors themselves, we believe "which" is the more suitable choice, and therefore, we have retained it without any alterations.
3) Use the past tense (The authors meticulously detail the process of distinguishing between various conditions in the diagnosis of this patient with AHEM).
Response: We appreciate your valuable comments. The sentence has been revised in the past tense to read as follows: "The authors meticulously detailed the process of distinguishing between various conditions in the diagnosis of this patient with AHEM."
4) Add this ref on the brain MRI appearance of MOGAD DOI: 10.1177/1971400917698856.
Response: Your valuable comments are greatly appreciated. In response to your suggestion, we have added the reference to the brain MRI appearance of MOGAD with DOI: 10.1177/1971400917698856.
- As the possibility of a hemorrhagic MOGAD wasn't proposed by Chen et al. I suggest the authors to better contextualize their hypothesis, by providing a quick background on anti-MOG disease, through a better explanation of the possible MRI appearance of the disease (I suggest a reference on this topic in the attached file) and diagnostic criteria.
Response: We appreciate your valuable comments and helpful suggestions. Based on your feedback, we have included additional information in the manuscript to provide better context and explanation. The added information regarding the MRI appearance of MOGAD is as follows: "Brain MRI findings in MOGAD typically show fluffy or poorly delineated T2 hyperintense lesions in various brain regions, including supratentorial and often infratentorial white matter, deep gray matter, middle cerebellar peduncle, large brainstem (pons and medulla), and confluent cortical areas. Contrast enhancement patterns may be nonspecific, with leptomeningeal enhancement around the brainstem and linear leptomeningeal enhancement in cases of cerebral cortical encephalitis. These lesions may partially or completely resolve over time, with infrequent silent lesion accrual and extremely rare residual T1 hypointense lesions, aiding in the diagnosis and evaluation of MOGAD.”
In addition, we've included information about the diagnostic criteria for MOGAD, as you suggested: "The diagnostic criteria for MOGAD consist of three main components: (A) Core clinical demyelinating events, including conditions such as optic neuritis, myelitis, ADEM, cerebral deficits, brainstem (B) positive MOG-IgG test results, which can be clear positive in serum without additional features, low positive in serum with supporting clinical or MRI features, or positive in CSF with supporting features and AQP4-IgG seronegative status; and (C) exclusion of better diagnoses, including multiple sclerosis. According to these diagnostic criteria, if this patient, who has core clinical demyelinating events of brainstem deficit and MRI features supporting brainstem syndrome, tests positive for MOG antibodies, a diagnosis of MOGAD can be made, regardless of the antibody titer being high or low."
We hope that these revisions provide the necessary background and context to better support the hypothesis and understanding of MOGAD in the manuscript.

Round 2
Reviewer 2 Report
I thank the authors for performing the requested changes